# Correlates of Commuter Cycling in Three Norwegian Counties

**DOI:** 10.3390/ijerph16224372

**Published:** 2019-11-08

**Authors:** Solveig Nordengen, Denise Christina Ruther, Amund Riiser, Lars Bo Andersen, Ane Solbraa

**Affiliations:** 1Faculty of Education, Arts and Sports, Department of Sport, Food and Natural Sciences, Western Norway University of Applied Sciences, Post box 133, 6851 Sogndal, Norway; amund.riiser@hvl.no (A.R.); lars.bo.andersen@hvl.no (L.B.A.); ane.solbraa@hvl.no (A.S.); 2Norwegian School of Sports Science, Oslo, Department of Sports Medicine, Post box 4014 Ullevål stadion, 0806 Oslo, Norway; 3Faculty of Engineering and Science, Department of Environmental Sciences, Western Norway University of Applied Sciences, Post box 133, 6851 Sogndal, Norway; denise.christina.ruther@hvl.no

**Keywords:** bicycle, public employees, active travel, active commuting, adults, GIS

## Abstract

Globally, there is an increasing challenge of physical inactivity and associated diseases. Commuter cycling is an everyday physical activity with great potential to increase the health status in a population. We aimed to evaluate the association of self-reported factors and objectively measured environmental factors in residence and along commuter routes and assessed the probability of being a commuter cyclist in Norway. Our study included respondents from a web-based survey in three Norwegian counties and we used a Geographic Information Systems (GIS) to evaluate the natural and built environment. Of the 1196 respondents, 488 were classified as commuter cyclists. Self-reported factors as having access to an e-bike (OR 5.99 [CI: 3.71–9.69]), being physically active (OR 2.56 [CI: 1.42–4.60]) and good self-rated health (OR 1.92 [CI: 1.20–3.07]) increased the probability of being a cyclist, while being overweight or obese (OR 0.71 [CI: 0.54–0.94]) reduced the probability. Environmental factors, such as high population density (OR 1.49 [CI: 1.05–2.12]) increased the probability, while higher slope (trend *p* = 0.020), total elevation along commuter route (trend *p* = 0.001), and >5 km between home and work (OR 0.17 [CI: 0.13–0.23]) decreased the probability of being a cyclist. In the present study, both self-reported and environmental factors were associated with being a cyclist. With the exception of being in good health, the characteristics of cyclists in Norway, a country with a low share of cyclists, seem to be similar to countries with a higher share of cyclists. With better knowledge about characteristics of cyclists, we may design better interventions and campaigns to increase the share of commuter cyclists.

## 1. Introduction

Globally, there is an increasing challenge of physical inactivity and several environmental factors are associated with physical activity (PA) levels [1]. Low levels of PA contribute to a higher risk of diseases [2]. The World Health Organization (WHO) recommends that adults be active at least 150 min/week in order to reduce the risk of non-communicable diseases (NCDs) [3]. Furthermore, it has been observed that any level of physical activity above sedentary is associated with a lower risk of mortality [4]. Commuter cycling is an everyday PA with great potential to increase the level of PA in the population.

Already in 2000, an association of lower risk of all-cause mortality among commuter cyclists was observed [5], and commuter cycling was later reported to be associated with a reduced risk of a number of illnesses, i.e., type 2 diabetes [6], cardiovascular disease [7], cancers [8], and obesity [9]. In two recent meta-analyses, cyclists compared to non-cyclists, had a lower body mass index (BMI), and were more physically fit [10,11]. Commuter cyclists have also been observed to be happier compared to car drivers [12]. Although cyclists have a higher risk of injuries, there is convincing evidence that the health benefit of cycling far outrun the risk of injury [13]. Due to all these positive associations of commuter cycling, it is important to understand the characteristics of those who are cyclists.

In the Netherlands, a country with a high share of commuter cyclists, cyclists live closer to work and are more physically active [14]. In Australia, commuter cyclists are more likely to be male, younger, and well-educated compared to non-cyclists [15]. For built (i.e., cycle infrastructure and connectivity) and natural environment (i.e., topography) the evidence of associations of share of commuter cycling is sparse. Previous studies have observed that distance [16], time to travel by bike relatively to time by car [16] and increased cycle infrastructure seem to increase the share of cyclists [16,17].

In countries with a generally low share of commuter cyclists, like Norway, we know less about which characteristics are associated with cycling. However, those owning an e-bike seem to be more likely to use their e-bike and travel longer distances compared to those with an ordinary bike [18].

Therefore, this study aims to describe the (a) self-reported characteristics of cyclists in a country with low levels of commuter cycling, and (b) the objectively measured environmental factors in areas around residence and along commuter routes associated with commuter cycling.

## 2. Materials and Methods

### 2.1. Sample

We invited all public sector employees in three Norwegian counties (Sogn og Fjordane, and Aust-Agder, and Vest-Agder (hereafter Agder). In general, Sogn og Fjordane is more hilly, wetter and windier than Agder. The study design, recruitment, and data collection have been described previously [18]. Briefly, during spring and autumn 2017, in total, 38,297 public sector employees got access to the web-based survey. In round one (Sogn og Fjordane), the questionnaire included questions about background, travel habits, local environment, bike access and use, sickness and injuries, health (RAND-12, and quality of life), physical activity, and confidence to other people. In the second round (Agder) the questionnaire was shortened and left out some questions in all the subgroups. Questions about confidence in other people and health were fully excluded. The estimated duration of completing the questionnaire was 30 and 15 min in the first and second round, respectively. The study was approved by the Regional Committees for Medical and Health Research Ethics with reference 2016/1897/REK vest. Entering the survey was defined as informed consent.

In total, 3540 (9.2% of the invited) individuals entered the survey. To be included in the analysis, dependent and independent variables needed to be reported. We included individuals between 18 and 72 years and excluded 17 participants due to extreme reports (age >72, height <1.3 m or >2.40 m, income >20,000,000 NOK, weight <44 kg or >200 kg). In total, 1196 individuals where included in the present study, see Figure 1. In the sub-analysis of distance cycled, 19 cases were excluded due to distance being >35 km from residence to work.

### 2.2. Being a Cyclist

Being a cyclist was defined by the Active Transport Norway-questionnaire [19]. In the present study, we only included the destination “work”. Test-retest reliability among adults cycling to work has previously been reported to be 0.92 (Spearman’s correlations) [19]. Those who reported one or more weekly trip(s) were classified as cyclists and the rest as non-cyclists. Distance to work was sampled by self-reported distance to work from residence.

### 2.3. Self-Reported Covariates

Self-reported age and perceived road safety were treated as continuous variables. Gender, type of cycle owned (e-bike or ordinary), ethnicity (Norwegian vs. non-Norwegian), self-rated health (SRH) (good or poor) and current tobacco (tobacco or non-tobacco) usage were coded binary. SRH was investigated by RAND-12′s first question. This question provides relevant health information and is a strong and dose-dependent predictor of mortality [20,21]. The question was recoded from “good” (good, very good and excellent) to “poor” (poor and fair) health status. Income, BMI, education, and self-reported PA [22] were coded as categorical variables. The Saltin and Grimby question of PA [22] has previously been used in a number of cohort studies assessing health status in the Nordic countries [5], and in a Norwegian representative population, where the question was validated against aerobic fitness (correlation coefficient was 0.18 and 0.39 for men and women, respectively) [23]. Through its use in cohort studies, this question has proven to be able to distinguish health and mortality between inactive and active respondents [5]. Income was classified as either 0–399,999 NOK, 400,000–799,999 NOK, or 800,000–19,999,999 NOK. BMI was classified according to WHO’s obesity classification [3]. Level of education was coded as <high school, <4 years university, and ≥4 years university. See Appendix A for more details.

### 2.4. Geographic Information Systems (GIS) Computed Covariates

Environmental factors were investigated in a GIS (ESRI ArcGIS PRO 2.3.3, Environmental Systems Research Institute, California, CA, USA). Participants’ home and work addresses (*n* = 1114) were geocoded using the address locator Environmental Systems Research Institute (ESRI) world geocode. This resulted in 1080 matched home addresses (97%), and 1053 work addresses (95 %), Figure 2a,c. Road network and shared-path network were imported from the Norwegian Public Road Administration toolbox NVDP-API. Meter of roads (European Road, State Road, County Road, Local Road, Private road, Logging road), and shared-use path were imported for Sogn og Fjordane (Figure 2b) and Agder (Figure 2d). The population was summarized at the district level. Districts were categorized by the number of persons living within the district into low (0–199), moderate (200–599), and high (>600) density groups (Figure 2a,c). To estimate the route between home and work (home–work pairs) we used the network analysis tool “routes”. This tool provides distance and travel time. For bike-route, the time-cost was estimated by calculating the time taken to travel the distance with an average speed of 15 km/h. Furthermore, we calculated the ratio between the time used when bicycling vs. driving, and the ratio between distances of the home–work route. Topography along routes for each of the original home–work-pairs was derived by cumulative absolute height gains (total elevation) and mean slopes from the Vbase data source. Elevation and slope were categorized into four groups based on percentile distribution to ensure a similar size of the groups. See Appendix A for details.

### 2.5. Statistics

#### 2.5.1. Cyclists vs. Non-Cyclists

An independent samples Mann–Whitney U test was used to investigate possible differences between cyclists and non-cyclists for non-normally distributed variables. Logistic regression was performed to assess the association between independent variables and being a cyclist. Model 1 contained 12 independent variables (age, distance, gender, income, health status, BMI, e-bike, education, migration, perceived traffic safety, tobacco and PA levels) from the questionnaire. The categorical variables were coded with ascending rank. The lowest group was used as a reference. Women and men were coded 0 and 1, respectively. Both bivariate and multivariate analyses were performed. Model 2 contained eight GIS-generated variables. As for model 1, categorical variables were coded with ascending rank. Stratified analyses were run for gender and counties (Sogn og Fjordane and Agder). See Appendix A for details.

#### 2.5.2. Distance Cycled

Correlates of self-reported distance to work (0–35 km) among cyclists were explored by linear regression. Distance to work was skewed (skewness = 2.07) and was therefore log-transformed by natural logarithm to ensure normal distribution (skewness ln(distance) = 0.25). The dependent variable was distance to work for cyclists, while the independent variables were all the variables for models 1 (questionnaire) and 2 (GIS variables). In total, 307 respondents were included. Stratified analyses were run for gender and counties (Sogn og Fjordane and Agder).

All analyses were run in IBM SPSS Statistics v. 25.0 (IBM Corp., Armonk, NY, USA). Descriptive analyses are presented as mean (SD) or median (min–max). Logistic regression is presented as odds ratio (OR) with a 95% confidence interval [CI], or with trend *p*-value for variables with more than two categories (education, income, and PA). The results of linear regression are presented as standardized beta (β), and *p*-value (*p*).

## 3. Results

Descriptive statistics are presented in Table 1. The multivariate model including survey data was able to distinguish between cyclists and non-cyclists, *p* < 0.001. The model explained 30% (Negelkerke R Square) of the variance of cycling behavior and correctly classified 74% of all cases. The multivariate model containing GIS data was able to distinguish between cyclists and non-cyclists *p* < 0.001. The model explained 14.9% of the variance of being a cyclist, and correctly classified 68.5% of all cases.

### 3.1. Being a Cyclist

Compared to non-cyclists, cyclists travelled significantly shorter (*p* < 0.001) (7.6 [10.7] vs. 21.1 [19.8] km) distances, perceived lower road safety (*p* < 0.001) (7.3 [2.2] vs. 6.6 [2.4]), and were slightly older (*p* = 0.043) (48.7 [10.6] vs. 47.4 [10.6] years). Those owning an e-bike or having an active lifestyle were six and two-fold more likely to be cyclists, respectively. Furthermore, higher level of education and good SRH were also associated with increased odds of being cyclist, whereas being overweight/obese reduced the odds of being a cyclist. Those living ≥5 km from work were unlikely to be cyclists. See Table 2 for details. The associations were similar for summer and winter, but SRH was more prominent during winter (OR 1.83 [1.15–2.93] vs. 2.42 [1.41–4.14]). Between the counties, SRH was significant for Sogn og Fjordane but not for Agder, while owning an e-bike was significant in Agder, but not Sogn og Fjordane. See Table A1 for details.

Among the environmental factors (GIS model, Table 3) all the factors were significantly associated with being a cyclist in the bivariate analysis. In the multivariate analysis of environmental factors, living in areas with higher population density and taking more time when cycling decreased the odds of being a cyclist.

The odds of being cyclist was similar for women and men when summer and winter were analysed combined. However, when gender was stratified per season, we observed that SRH during summer was more strongly associated among men (OR 2.54 [1.23–5.23]) than among women (OR 1.45 [0.77–2.72]), while level of PA was more strongly associated among women (*p* for trend = 0.010), compared to men (*p* for trend =0.179). For winter, e-bike increased the chances of being a cyclist more for women than it did for men (OR 7.55 [3.99–14.03] vs. 3.61 [1.73–7.54]). For environmental factors (model 2), there were similar results for men and women and between counties. See Table A2 for the results of environmental factors at the county level.

Distance from residence to work was observed to correlate with frequency of cycling, and thus, the average weekly distance cycled. Most of those who were cyclist had a short distance (0.1–20 km) to travel to work and thus had a low to moderate dose (10–60 km) of distance cycled in an average week. It seems like those living 5–10 km from work cycled more often than others and gained a larger weekly average compared to both shorter and longer distances.

### 3.2. Distance Cycled

Among cyclists, we observed that distance cycled was associated with being male, a lower level of perceived road safety, having a more beneficial ratio of shared-use path/roads at home buffer, and a low total elevation and mean slope. See Table 4 for details. The negative association between perceived road safety and distance cycled indicates that those cycling shorter distances are more affected by road safety compared to those cycling longer distances. This was investigated further and a significant correlation was found (chi-square = 0.013) for high perceived road safety reported among those cycling short distances (1–2 km). For summer and winter separately, associations were similar, with the expectations of winter SRH, which was significantly associated with longer cycling distances (β = 0.13 *p* = 0.031).

## 4. Discussion

The present study aims to describe the association between commuting by bicycle, self-reported characteristics and objectively measured environmental factors. Among the 1196 included participants, 488 were cyclists. Owning an e-bike, being active, and with good health increased the probability of being a cyclist by almost six-, three- and two-fold-larger odds, respectively, compared to non-cyclists. On the other hand, living >5 km from work reduced the probability of being a cyclist by 83%, and being overweight or obese reduced the probability by 29%. For the environmental factors, living in more populated areas increased the odds by almost 50%, while having a total elevation of more than 133 m reduced the odds of being a cyclist by almost 50%.

In the self-reported data, we observed that men were more likely to be cyclists. This is a similar finding to countries with a higher share of cyclists [15,24,25]. Owning an e-bike gave a six-fold increase in the probability of being a cyclist and has been discussed elsewhere [18]. Furthermore, we observed that those with higher education were more often cyclists. This is also in accordance with observations from Australia [15], Europe [24], and North America [25]. In accordance with previous findings in Europe [14], those being categorized as physically active were up to three times more likely to be cyclists. This indicates that those who cycled for transportation may often be engaged in other forms of physical activity. Interestingly, there was an almost two-fold likelihood of being a cyclist among those reporting good health status. This is in contrast to observations in Brussels where SRH was not related to commuter cycling [24]. In both studies, the proportions of respondents with good health were high (~90%). This may indicate that health status is one of the few factors that differs in a country with a low share of cyclists compared to countries with a higher share of cyclists. In the study, those who were cyclists may be a selected group of individuals who are highly educated, physically active, normally weighted and in good health. However, a lower incidence of CVD and death was observed in a meta-analysis of more than 1 million individuals [10] even when most of the included studies were adjusted for physical activity and education.

Cyclists travelled one third of the distance compared to non-cyclists, whilst those living >5 km from work were rarely cyclists. Barton at al. [26] observed that the distance between locations had to be short (500–2500 m) for active travel in the UK. Our findings confirm that commuter cycling is more typical when the commuting distance is relatively short (<5 km), albeit twice as long as the UK findings [26]. In Norway, the average travel distance between home and work is 16.3 km, and only seven percent are undertaken by bike [27]. Independent of mode of transportation, 39% of all journeys are <5 km [27]. In our study, 39% lived less than 5 km from work. The included respondents thus seem to be fairly representative for the whole of Norway concerning living less than 5 km from work. It may be that Norwegian commuter cyclists are willing to travel longer distances compared to those in the UK. However, the willingness to travel longer might also be affected by the exclusive focus on cycling in our study, whereas Barton et al. [26] considered active travel in general, including walking and cycling. In the UK, walking is twice as common as cycling [28]. Interestingly, short distances have been reported to be of greater importance than safety when it comes to choices of route among cyclists [29]. This may be why we observed that longer distances were associated with lower perceived safety, as the cyclist may choose a more unsafe route to reduce the travel distance.

Another important observation in our study, the positive association between population density and probability of being a cyclist, is in accordance with previous reports of commuter cycling [16], active travel [26], and a higher level of physical activity [1]. In more populated areas, the distance between home and work is often shorter [16]. If there is a 5 km threshold for trips to be conducted by bike, it follows that there is a higher potential for trips to be made by bike in such areas. However, in Norway, large areas have scattered settlements (Figure 2a,c), and this may be why Norwegian cyclists cycle longer distances compared to the abovementioned observations from the UK. The scattered settlement in Norway is also a factor that cannot easily be changed, and may be one of the main reasons why the share of cyclist has not increased [27] despite raised focus in the Norwegian transport plan and cycle strategy [30].

When the travel time by bike is shorter relative to time travelled by car, more people are likely to cycle. This is in accordance with findings from British cities and towns [16] and other interventions on bicycle infrastructure [29]. The present ratio is based on distance between home and work, and the route is estimated by the GIS-tool routes, choosing the most likely route for bike and car. We used an average speed of 15 km/h for cyclists, while car was set to default by the tool. This means that the commuting route may differ between the one by bike compared to the one taken by car. Interestingly, we did observe a positive association for the ratios of shared-use path/roads at home, but not for either car or bike junctions along routes. This is in contrast to the observations by Cervero et al. [16], who observed that increased connectivity increased the share of cyclists. However, our ratio included shared-use path, not exclusively cycle infrastructure. Exclusive cycle infrastructure in Norway is sparse and much rarer than shared-use paths (Figure 2b,d).

In the bivariate results of the model containing GIS-generated variables, we observed a significant negative trend for both mean slope and elevation on commuting route (*p* < 0.001–0.042). Only for elevation along the route, the trend remained significant in the multivariate model. This indicated that a commuter who travelled with a total elevation of 133 vertical meters was 57%–63% less likely to commute by bike. Our finding is in accordance with previous observations where vertical displacement along commuter route was negatively associated with the probability of being a cyclist [16,31].

The present study cannot conclude on causality, but there seems to be a relationship between the level of PA, population density and the ratio of shared-use path and roads at home and the distance cycled. Our findings are in accordance with observations for Vancouver city where built and natural environments were associated with the share of cyclists [31]. Thus, it seems likely that the built and natural environment affect the level of commuter cycling.

### 4.1. Strengths and Limitations

The use of geographical data provided a direct measure of the environment in the investigated area. Together with the self-reported information, we were able to see a large picture of which factors were associated with commuter cycling. This is important information for politicians, policy makers and city planners.

The main strength of this study is the combination of a relatively large sample and the inclusion of GIS-measures of population and the environment. The sample is from two large geographical areas of Norway (Figure 2a–d). The broad requirement strategy seems to have succeeded for geographical distribution, i.e., travel distance to work, but the sample is more active, less obese, more highly educated and has a higher income than the general population in Norway. However, our aim was to describe the characteristics of cyclists, which is possible to do based on a sample consisting of 41% cyclists. For the commuting route, we observed that mean slope in the >75% percentile data gave non-logic results, where the highest group of slope increased the odds of being a cyclist. This is likely due to errors in the dataset. However, we chose to include total elevation change and mean slope derived from the TIN in the analysis since topography is likely to be one of the main factors associated with cycling [31]. Unfortunately, we had a very low response rate of only 3%. However, analysis of associations is quite robust to selection bias, and response rate is therefore of less importance. The sample of this survey was selected and not representative of the general population. However, associations between cycling and other parameters may still be valid. It is usually seen that physical activity has a preventive effect in all groups independent of age, sex and other parameters. Similarly, it is likely that the responders may choose or not choose to cycle, similarly to the total population.

### 4.2. Interpretation

We interpreted the result to identify and understand both personal, natural and built environment factors. With more knowledge about the characteristics of cyclists, we may design better interventions and campaigns to increase the share of commuter cyclists. The present study identified a number of factors, such as population density, elevation along commuting route, level of PA and gender that were significantly associated with commuter cycling. There seems to be no single factor affecting people’s choice of transportation mode [1,16]. However, adaptions in the built environment in areas with a high population density and a likely lower distance between home and work may increase the share of cyclists.

## 5. Conclusions

In the present study, both self-reported and environmental factors were associated with odds of being a cyclist. Owing an e-bike, being active and in good health increased the odds of being a cyclist, while living more than 5 km from work and being overweight or obese reduced the probability of being a cyclist. With the exception of being in good health, the characteristics of cyclists in a county with a low share of cyclists seems to be similar to countries with a higher share of cyclists. Adaption of the built environment in areas with a high population density and shorter distances between home and work may increase the share of cyclists. Future studies should investigate which changes in the environment may increase the share of cyclists and aim to better understand hampers for changing transit from car to bike.

## Figures and Tables

**Figure 1 ijerph-16-04372-f001:**
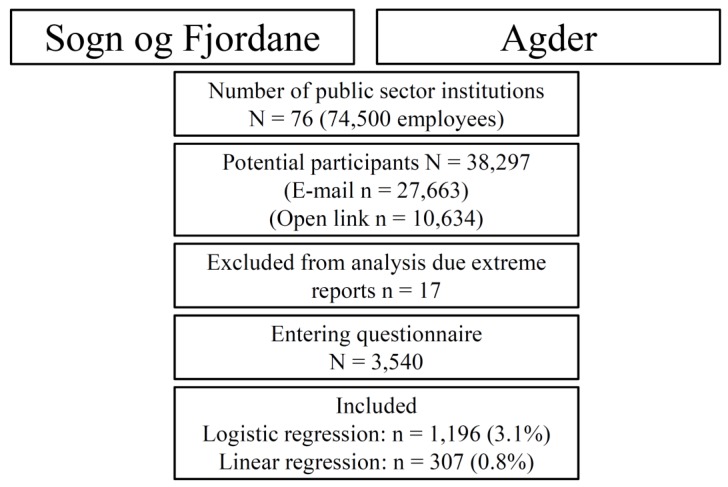
Flowchart and inclusion process of the Førde Active Transport study.

**Figure 2 ijerph-16-04372-f002:**
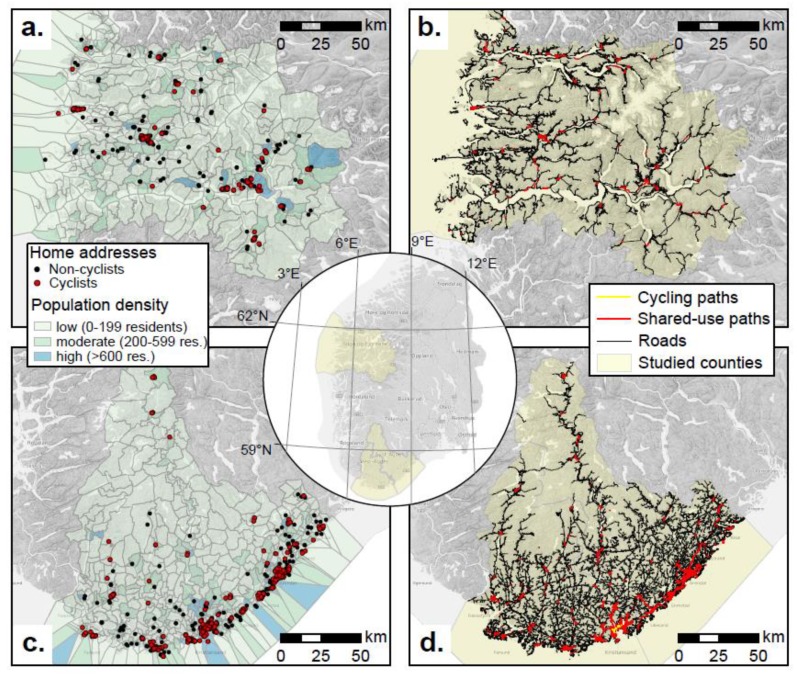
Geographic Information Systems (GIS)-derived information. (**a**) Population density and location of home addresses in Sogn og Fjordane; (**b**) Roads, cycling paths and shared-use paths in Sogn og Fjordane; (**c**) Population density and location of home addresses in Agder; (**d**) Roads, cycling paths and shared-use paths in Agder.

**Table 1 ijerph-16-04372-t001:** Descriptive table of characteristics of participants, *n* = 1196.

Characteristics	Sogn og Fjordane and Agder		County	
	Cyclists	Total	Sogn og Fjordane	Agder
			Total (% Cyclist)	Total (% Cyclist)
*n*	488	1196	441 (35%)	755 (41%)
Distance (*n* = 1196)				
0.1–5.0 km	301	467	183 (62%)	284 (66%)
5.0–145 km	187	729	258 (16%)	471 (31%)
Age (median (min-max))	49 (19–70)	49 (72–19)	48 (23–72) ^a^ 49 (67–24^) b^	49 (19–70) ^a^ 49 (19–70) ^b^
Gender (*n*)				
men	204	468	155 (38%)	313 (46%)
women	284	728	286 (33%)	442 (42%)
Income (*n*)				
0–399,999 NOK	69	266	92 (30%)	174 (40%)
400,000–799,999 NOK	371	868	321 (38%)	547 (46%)
800,000–19,999,999 NOK	21	62	28 (25%)	34 (41%)
Self-reported health status * (*n*)				
Poor	38	138	44 (18%)	94 (32%)
Good	450	1058	397 (37%)	661 (46%)
BMI (*n*)				
Underweight or normal weight	282	627	246 (40%)	381 (48%)
Pre-obesity or Obesity class 1–3	206	569	195 (29%)	374 (40%)
Tobacco (*n*) *				
Non-tobacco	484	1188	438 (35%)	750 (44%)
Any usage of snuff or tobacco	4	8	3 (66%)	5 (40%)
Cycle type (*n*)				
other	408	1083	432 (39%)	651 (39%)
e-bike	80	113	9 (33%)	104 (74%)
Ethnicity (*n*)				
Self and parents born in Norway	428	1080	401 (34%)	679 (43%)
Self or parents not born in Norway	60	116	40 (48%)	76 (54%)
Education (*n*)				
<high school	50	157	50 (30%)	107 (33%)
University <4 years	98	273	103 (29%)	170 (40%)
University ≥4 years	340	766	288 (38%)	478 (48%)
Road safety (median (min-max))	8 (1–10)	8 (1–10)	7 (1–10) ^a^ 8 (1–10) ^b^	8 (1–10) ^a^ 8 (1–10) ^b^
PA level ** (*n*)				
inactive	20	95	40 (22%)	55 (20%)
Activity class 1	246	602	209 (33%)	393 (45%)
Activity class 2 or 3	222	499	192 (40%)	307 (47%)
Population density (*n* = 730)				
1	94	230	55 (18%)	175 (48%)
2	96	241	86 (38%)	155 (41%)
3	129	259	143 (43%)	116 (58%)
Mean slope route *n* = 730				
<25% 0–3.8%	83	170	94 (43%)	76 (57%)
25–50%, 3.8–5.6%	71	187	119 (34%)	68 (44%)
50–75%, 5.6–14.0%	68	179	49 (37%)	130 (38%)
>75%, >14.0%	97	194	22 (27%)	172 (53%)
Sum elevation home-work-home *n* = 730				
<25%, 0–132.7 m	119	172	69 (70%)	103 (69%)
25–50%, 132.7–555.9 m	72	188	75 (29%)	113 (44%)
50–75%, 555.9–1509.6 m	66	194	50 (16%)	144 (40%)
>75%, >1509.6 m	62	176	90 (30%)	86 (41%)

* Tobacco included both snuff and smoke. Non-tobacco included those who are non-users. ** Based on the four activity categories by Saltin and Grimby [22]: “Almost completely inactive: reading, TV watching, movies, etc.” [inactive], “Some physical activity during at least 4 h per week, riding a bicycle or walk to work, walking or skiing with the family, gardening’’ [1], “Regular activity, such as heavy gardening, running, calisthenics, tennis, etc.” and “Regular hard physical training for competition in running events, soccer, racing. European handball, etc. several times per week.” [2]. ^a^ cyclists; ^b^ non-cyclists.

**Table 2 ijerph-16-04372-t002:** Likelihood of being a cyclist, survey data, *n* = 1196. Presented as bivariate and multivariate analyses. Significant associations are written in bold.

Characteristics	Bivariate	Multivariate
	All Seasons OR (95% CI)	All Seasons OR (95% CI)
Age	**1.01 (1.00–1.02); 0.043**	1.01 (0.99–1.02); 0.100
>5 km vs. <5 km distance	**0.19 (0.15–0.25); <0.001**	**0.17 (0.13–0.23); <0.001**
Gender (women vs. men)	1.21 (0.95–1.53); 0.116	**1.45 (1.09–1.92); 0.010**
Income	**Trend *p* = 0.082**	Trend *p* = 0.086
Income (0–399.999NOK)	Ref.	Ref.
Income (4–799.999)	1.32 (1.00–1.76); 0.054	1.09 (0.77–1.53); 0.632
Income (>800.000)	0.91 (0.51–1.63); 0.743	0.54 (0.28–1.067); 0.077
SRH poor vs. good	**1.95 (1.31–2.89); 0.001**	**1.92 (1.20–3.07); 0.007**
Normal weight vs. Pre-obesity or Obesity class 1–3	**0.69 (0.55–0.88); 0.002**	**0.71 (0.54–0.94); 0.017**
E-bike	**4.01 (2.63–6.13); <0.001**	5.99 (3.71–9.69); <0.001
Education	**Trend *p* = 0.003**	**Trend *p* = 0.023**
Education< high school	Ref.	Ref.
<4 years university	1.20 (0.79–1.81); 0.395	1.33 (0.82–0.2.15); 0.246
≥4 year university	**1.71 (1.19–2.46); 0.004**	**1.75 (1.14–2.70); 0.011**
Ethnicity	**1.63 (1.11–2.40); 0.012**	**1.69 (1.08–2.64); 0.021**
Perceived Road safety	**1.13 (1.08–1.19); <0.001**	1.05 (0.99–1.12); 0.081
Tobacco	1.46 (0.36–5.84); 0.597	0.69 (0.12–4.02); 0.675
Activity class *	**Trend *p* < 0.001**	**Trend *p* = 0.002**
Activity class 1	Ref.	Ref.
Activity class 2	**2.59 (1.54–4.36); <0.001**	**2.56 (1.42–4.60); 0.002**
Activity class 3	**3.01 (1.78–5.08); <0.001**	**2.90 (1.60–5.26); <0.001**

* Based on the four activity categories by Saltin and Grimby [22], “Almost completely inactive: reading, TV watching, movies, etc.” [inactive], “Some physical activity during at least 4 h per week, riding a bicycle or walk to work, walking or skiing with the family, gardening’’ [1], “Regular activity, such as heavy gardening, running, calisthenics, tennis, etc.” [2] and “Regular hard physical training for competition in running events, soccer, racing. European handball, etc. several times per week.” [3]. SRH, self-rated health status.

**Table 3 ijerph-16-04372-t003:** Likelihood of being a cyclist. Environmental factors (GIS data). *n* = 1009. Presented as bivariate and multivariate analyses. Significant associations are written in bold.

	Bivariate	Multivariate
	All Seasons OR (91% CI); *p*	All Seasons OR (91% CI); *p*
*N*	1009	1009
**500 m home buffer**		
Ratio shared-path/road buffer home	**3.62 (1.29–10.19); 0.015**	1.79 (0.42–7.69); 0.435
Car junction home	**1.01 (1.00–1.01); <0.001**	1.00 (0.99–1.01); 0.598
Bike junction home	**1.00 (1.00–1.01); <0.001**	1.00 (0.99–1.01); 0.869
Population density home	**Trend *p* < 0.001**	Trend *p* = 0.058
Low (0–199 persons)	Ref.	Ref.
Moderate (200–599 persons)	1.11 (0.80–1.54); 0.551	1.09 (0.77–1.55); 0.626
High (<600 persons)	**1.81 (1.32–2.47); <0.001**	**1.49 (1.05–2.12); 0.026**
**Route**		
Ratio minutes home-work bike */car route	**0.55 (0.47–0.63): <0.001**	**0.72 (0.56–0.93); 0.013**
Ratio meter bike/car route	**0.04 (0.01–0.18); <0.001**	0.83 (0.15–4.65); 0.831
Percentiles of mean slope route	**Trend *p* = 0.042**	**Trend *p* = 0.020**
<25% 0–3.8%	Ref.	Ref.
25–50%, 3.8–5.6%	0.76 (0.53–1.10); 0.143	0.91 (0.60–1.36); 0.636
50–75%, 5.6–14.0%	**0.60 (0.41–0.86); 0.006**	0.75 (0.49–1.13); 0.162
>75%, >14.0%	0.87 (0.61–1.24); 0.439	1.44 (0.91–2.28); 0.125
Percentiles for elevation t/r route	**Trend *p* < 0.001**	**Trend *p* = 0.001**
<25%, 0–132.7 m	Ref.	Ref.
25–50%, 132.7–555.9 m	**0.32 (0.22–0.46); <0.001**	**0.43 (0.28–0.67); <0.001**
50–75%, 555.9–1509.6 m	**0.26 (0.18–0.37); <0.001**	**0.37 (0.21–0.64); <0.001**
>75%, >1509.6 m	**0.27 (0.19–0.39); <0.001**	**0.44 (0.23–0.84); 0.013**

* estimated 15 km/h.

**Table 4 ijerph-16-04372-t004:** Linear regression of distance * cycled. *n* = 307. Significant associations are written in bold.

	All Seasons	
	β	*p*-Value
**Survey**		
Age	0.039	0.454
Gender (women vs. man)	0.109	**0.035**
Income (ascending)	−0.038	0.469
SRH (poor vs. good)	0.087	0.100
Normal weight vs. overweight/obesity	−0.054	0.312
E-bike (regular vs. e-bike)	0.041	0.443
Years of education (ascending)	0.014	0.794
Perceived road safety (ascending)	**−0.220**	**<0.001**
Ethnicity (ethnic Norwegian vs. not ethnic Norwegian)	0.017	0.744
PA level (ascending)	0.046	0.388
**GIS**		
500 m home buffer		
Population density home	−0.025	0.637
Bike junction home	0.063	0.700
Car junction home	−0.338	0.062
Ratio shared-path/road buffer home	**0.184**	**0.007**
Route		
Ratio minutes home-work bike **/car route	0.035	0.713
Ratio meter bike/car route	0.071	0.272
Percentiles of mean slope route	**0.188**	**0.004**
Percentiles for elevation t/r route	**0.232**	**0.013**

* Distance is log-transformed; ** Estimated 15 km/h; β, Standardized beta; SRH, self-rated health status; PA, physical activity; GIS, geographic information systems.

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
