# Peer review of "Correlates of Commuter Cycling in Three Norwegian Counties"

_ijerph, 2019, doi:10.3390/ijerph16224372_

Round 1

Reviewer 1 Report

The paper contains valuable information and is well-structured. Several minor issues require the attention of authors.

The authors can give the date of the survey, as well as the average duration of completing the questionnaire. Discussion and interpretation of the survey results can be presented beside the tables of the results.   Important findings can be summarized in the conclusion sector. Potential future words can also be given in the conclusion sector.

Reviewer 2 Report

This paper investigated the commuter cycling behavior and its relationship with topography, distance as well as e-bike ownership. The association of self-reported factors and objectively measured environmental factors in residence and along commuter routes were evaluated. The entire paper is in a rather good logic, with good innovation in content. The reviewer only has minor reviewing comments as follows:

1. The title is not appropriate “Commuter cycling is associated with topography, distance, and possessing e-bikes”, which may be changed.

2. Some indexes provided in Abstract, such as “(OR 5.99 [CI: 3.71-9.69])” were not understandable, which may be removed. In fact, the abstract may be re-organized to include background, objective, method, results, conclusion and possible applications. Some details may be included in a summarized way, particularly for the results and conclusions.

3. The self-reported or survey is actually more related to “Self-revealed” or “Self-exposed” processes. The objective and theoretically background may be provided. Additionally references may be obtained from:

Sun and Elefteriadou, 2011. Lane changing behavior on urban streets: A focus group based study, Applied Ergonomics: Human Factors in Technology and Society, 42(5), pp. 682-691.

4. In page 1, lines 37-38, “… less than an hour daily PA of moderate intensity may protect against mortality risk associated with inactivity” the sentence is not understandable. “Less than an hour daily PA” is better than “more than 1 hour”?

5. The flowchart in Figure 1 may be modified into a more formal one.

6. Figure 2 a-d are not readable. Why Figure 2.a only provides home address, with no work address?

7. In page 4, lines 123-124,“Logistic regression was performed to assess the association between independent variables and being cyclists.”

8. In line 218, the authors mentioned that “… having a total elevation of more than 133m …” why the value 133 was used here?

9. How did the weather and the availability of bicycle affect the activities? Particularly, the sharing bikes and E-scooter are popular in many cities world widely.

10. In page 10, lines 290-300, the authors tried to compare the DSM, DTM and TIN, and explain the reasons in choosing the TIN model, which is actually not necessary.

11. The conclusion section is too short which should be expanded to include the conclusions and potential future studies of the research.

12. Additional references related to public transit, cyclists and pedestrians may be included.

Reviewer 3 Report

The topic is interesting and relevant to the Norwegian country, as there is no previous information about characteristics of its cyclists. The fact of including GIS in the study provides a plus of novelty. However, there are some important issues that should be improved.

Especially, I shall raise my concern about the methods. As this is a descriptive study, it is essential to calculate a sample size according to a selected accuracy. The authors do not present it in the manuscript; instead, they accept the final number of participants who answered the survey. The response rate, besides, seems to be very low, as the authors themselves recognize in Strenghts and Limitations section. Perhaps it would be useful to implement the CATI approach (Computer-assisted Telephone Interviewing)

Also, the sample collection poses another problem. The answer of the survey is completely voluntary, so it could introduce a selection bias (we have no information about the characteristics of the people who preferred not to answer, and they could be very different). The authors seem to dismiss this bias in page 10 (line 302), but I'm afraid I can't agree with them. I'd like to read a more exhaustive reasoning.

According to Material and Methods section, I see unnecessary the extended methods appendix -which makes the reading uncomfortable. Instead, I suggest to write the entire Methods in the manuscript.

Results section and Tables seem a bit confusing to me. Please watch the footnotes corresponding to asteriks in Tables 1 and 2 -are they OK? Also, in Table 1, I cannot find the key for "a" and "b" (superscript). The paragraph following Table 3 (lines 184 to 191) makes reference to analysis stratified by season, but these data are not shown anywhere (although the authors seems to suggest that they are in Table 3, in Appendix A).

The objective of the study does not appear in the abstract.

Author Response

The topic is interesting and relevant to the Norwegian country, as there is no previous information about characteristics of its cyclists. The fact of including GIS in the study provides a plus of novelty. However, there are some important issues that should be improved.

R: Especially, I shall raise my concern about the methods. As this is a descriptive study, it is essential to calculate a sample size according to a selected accuracy. The authors do not present it in the manuscript; instead, they accept the final number of participants who answered the survey. The response rate, besides, seems to be very low, as the authors themselves recognize in Strenghts and Limitations section. Perhaps it would be useful to implement the CATI approach (Computer-assisted Telephone Interviewing).

A: Thank you for highlighting this important issue. We agree that there is a low response rate and that it may be biased as the survey was voluntary. Due to the low response rate we discussed if the survey was worthy publishing as we are aware of the bias and uncertainty. However, when including the data of the environment and keeping the focus on the cyclists, we argue that this is of interest for the field. Due to lack of resources (both time and money), we do not have the possibility to implement the CATI approach, even though it may have given important additional information.

R: Also, the sample collection poses another problem. The answer of the survey is completely voluntary, so it could introduce a selection bias (we have no information about the characteristics of the people who preferred not to answer, and they could be very different). The authors seem to dismiss this bias in page 10 (line 302), but I'm afraid I can't agree with them. I'd like to read a more exhaustive reasoning.

A: It is clear that the sample is selected and not representative for the general population. However, associations between cycling and other parameters may still be valid. It is usually seen that physical activity has a preventive effect in all groups independent of age, sex and other parameters. Similarly, it is likely the selected group which chose to answer the questionnaire may choose or not choose to cycle in a similar way as the total population.

We have added our reasoning in the end of the strength and limitation-section.

R: According to Material and Methods section, I see unnecessary the extended methods appendix -which makes the reading uncomfortable. Instead, I suggest to write the entire Methods in the manuscript.

A: Thank you for this comment. The method have an extended part to make the information is available to the reader with in-depth knowledge and interest in the methods used. If we include the extended methods in the manuscript, we are afraid the manuscript will be too long and unnecessary complicated for the common reader. Therefore, we like to keep the extended methods appendix. In case you still like us to merge the Methods from the appendix with the manuscript, we will accommodate this.

R: Results section and Tables seem a bit confusing to me. Please watch the footnotes corresponding to asteriks in Tables 1 and 2 -are they OK? Also, in Table 1, I cannot find the key for "a" and "b" (superscript).

A: The footnotes and asterisks have been edited. May you please clarify what is confusing with Table 1 and 2?

R: The paragraph following Table 3 (lines 184 to 191) makes reference to analysis stratified by season, but these data are not shown anywhere (although the authors seems to suggest that they are in Table 3, in Appendix A).

A: Table 3 in appendix A describes environmental factors at county level. This has now been clarified in the manuscript.

R: The objective of the study does not appear in the abstract.

A: The objective is now included in the abstract
